PGSXplorer: an integrated nextflow pipeline for comprehensive quality control and polygenic score model development

Yaraş Tutku 1 2
http://orcid.org/0000-0002-0158-2693 Oktay Yavuz 1 2 3
http://orcid.org/0000-0001-6706-1375 Karakülah Gökhan 1 2 gokhan.karakulah@deu.edu.tr
1 İzmir Biomedicine and Genome Center , İzmir , Turkey
2 İzmir International Biomedicine and Genome Institute, Dokuz Eylül University , İzmir , Turkey
3 Department of Medical Biology, Faculty of Medicine, Dokuz Eylül University , İzmir , Turkey
Bakir-Gungor Burcu
Electronic publication date: 2025 Feb 12
Publication date: 2025
Volume: 13
Electronic Location ID: e18973
Received 2024 Nov 1; Accepted 2025 Jan 21
Copyright: © 2025 Yaraş et al.
Copyright year: 2025
Copyright holder: Yaraş et al.
License: This is an open access article distributed under the terms of the Creative Commons Attribution License, which permits unrestricted use, distribution, reproduction and adaptation in any medium and for any purpose provided that it is properly attributed. For attribution, the original author(s), title, publication source (PeerJ) and either DOI or URL of the article must be cited.
License URL: https://creativecommons.org/licenses/by/4.0/

Keywords: Polygenic score, PGS, Polygenic risk score, PRS, Quality control, Nextflow, GWAS, Pipeline

Funding: The authors received no funding for this work

==============================
The rapid development of next-generation sequencing technologies and genomic data sharing initiatives during the post-Human Genome Project-era has catalyzed major advances in individualized medicine research. Genome-wide association studies (GWAS) have become a cornerstone of efforts towards understanding the genetic basis of complex diseases, leading to the development of polygenic scores (PGS). Despite their immense potential, the scarcity of standardized PGS development pipelines limits widespread adoption of PGS. Herein, we introduce PGSXplorer, a comprehensive Nextflow DSL2 pipeline that enables quality control of genomic data and automates the phasing, imputation, and construction of PGS models using reference GWAS data. PGSXplorer integrates various PGS development tools such as PLINK, PRSice-2, LD-Pred2, Lassosum2, MegaPRS, SBayesR-C, PRS-CSx and MUSSEL, improving the generalizability of PGS through multi-origin data integration. Tested with synthetic datasets, our fully Docker-encapsulated tool has demonstrated scalability and effectiveness for both single- and multi-population analyses. Continuously updated as an open-source tool, PGSXplorer is freely available with user tutorials at https://github.com/tutkuyaras/PGSXplorer, making it a valuable resource for advancing precision medicine in genetic research.

Introduction

In the post-Human Genome Project era, genomic data has become a cornerstone of personalized medicine and health research. The availability of technologies capable of generating vast genomic datasets has accelerated advancements in genome-wide association studies (GWAS) and other genomic analyses, providing insights into the genetic basis of complex diseases (Kim et al., 2012; Shi & Wu, 2017). The National Institutes of Health’s Genomic Data Sharing Policy has facilitated the dissemination of these datasets, improved collaboration and supported precision medicine initiatives (Shi & Wu, 2017). Central to this progress is genomic data quality control (QC), which relies on filtering and preprocessing genetic data to meet the stringent demands of high-throughput analyses and aims to ensure the accuracy and reliability of findings (Chang et al., 2015; Gondro, Porto-Neto & Lee, 2014).

One of the most significant advances arising from the GWAS approach is the creation of polygenic scores (also known as polygenic risk scores, PRS or PGS), which estimate an individual’s genetic predisposition to traits or diseases by aggregating the effects of multiple genetic variants. Calculated from genome-wide genotypes and their weights, which are determined by effect sizes from GWAS, PGS provide a single-value estimate of genetic propensity. These scores have shown great promise in predicting the risk of complex diseases like Alzheimer’s (Mantyh et al., 2023), Parkinson’s (Wen et al., 2023), cardiovascular diseases (e.g., coronary artery disease, atrial fibrillation) (Elliott et al., 2020; Kavousi & Ellinor, 2023), prostate cancer (Schaffer et al., 2023) among others. By improving predictive accuracy compared to earlier genetic tools, PGS support individualized medicine by offering clinicians insights that can guide interventions.

Recent advances in PGS computation have highlighted both challenges and opportunities in integrating diverse populations into genetic research. A major limitation in the field stems from the overrepresentation of individuals of European ancestry in GWAS datasets, which reduces the transferability and clinical utility of PGS for non-European populations (Fahed et al., 2020). To address these disparities, researchers have focused on developing population-specific PGS and adopting multidisciplinary approaches to deliver equitable and generalizable genetic risk estimates (Page et al., 2022; Smith et al., 2023). Emerging trends further emphasize the need to evaluate the utility of PGS across a broad spectrum of diseases, including cancer and coronary heart disease, while cautioning against the risks of overgeneralizing population-level data into individual risk predictions (Khan et al., 2023). However, the calculation and interpretation of PGS involve computationally demanding tasks, including managing large datasets and addressing population genetics’ complexities, necessitating efficient algorithms and QC protocols (Choi et al., 2020; Pain et al., 2020).

Automated genomic workflows, particularly those based on platforms like Nextflow, address these challenges by providing scalable, reproducible pipelines that ensure accurate data processing (Schulz et al., 2016). In this study, we developed PGSXplorer, a comprehensive Nextflow DSL2 pipeline that integrates QC steps and multiple PGS development algorithms to streamline the analysis of GWAS data. Our proposed pipeline is fully Dockerized, enhancing its portability, reproducibility, and usability across different platforms, while adhering to FAIR (Findability, Accessibility, Interoperability, and Reusability) principles (Tommaso et al., 2017). PGSXplorer employs a suite of methods to optimize PGS modeling, integrating well-established PGS algorithms. Our tool uniquely supports multi-ancestry analyses, notably enhancing PGS accuracy and generalizability across diverse populations. By incorporating genetic diversity across ancestries, it enables the creation of robust and inclusive PGS models. Additionally, automating processes such as genotype assignment, phasing, imputation, data filtering, and model construction makes the genomic workflow more efficient, positioning PGSXplorer as a valuable tool for advancing PGS research and facilitating large-scale genomic studies.

Materials AND methods

Pipeline development

PGSXplorer was developed using Nextflow (v24.04.3) (Tommaso et al., 2017) and the DSL2 language and executed within a Docker container. Dockerization was employed to ensure a consistent and reproducible environment through encapsulation of all dependencies and software required for the analysis. A suite of specialized tools for PGS model development were integrated into the pipeline, including PLINK (v1.9) (Chang et al., 2015), PRSice-2 (Choi & O’Reilly, 2019), LD-Pred2 (Privé, Arbel & Vilhjálmsson, 2020), Lassosum2 (Privé et al., 2021), MegaPRS (Zhang et al., 2021), SBayesR-C (Zheng et al., 2024), PRS-CSx (Ruan et al., 2022), and MUSSEL (Jin et al., 2023).

QC workflow of the pipeline

The QC workflow of our pipeline begins with the initial GWAS QC. This module automates key QC processes for GWAS summary statistics to ensure data integrity and reliability in downstream analyses. The QC module performs the following steps: MAF filtering: Variants with MAF values below 0.01 are excluded to remove rare variants that may cause noise in the analysis.

INFO filtering: Variants with INFO (imputation score) values below 0.8 are removed to enhance genotyping accuracy and minimize potential bias.

Duplicate SNP removal: Duplicate SNPs are systematically identified and removed to avoid redundant or conflicting data.

Following these steps, the filtered GWAS summary statistics are reconstructed and prepared for subsequent stages of the workflow. These QC procedures follow standard guidelines outlined by a previous study (Choi, Mak & O’Reilly, 2020).

Our pipeline was also designed to optimize the QC of the user-provided genotype data through a series of automated steps: (i) filtering missing SNPs, (ii) filtering missing individuals, (iii) minor allele frequency (MAF) filtering, (iv) Hardy–Weinberg equilibrium (HWE) filtering, (v) relatedness check, (vi) heterozygosity assessment, and (vii) removal of duplicate SNPs. These QC steps are streamlined and are executed using PLINK (v1.9) with the parameters --geno, --mind, --maf, --hwe, --rel-cutoff, and --het (Chang et al., 2015). Additionally, customized scripts written in R (v4.1.0) were employed to visualize heterozygosity, HWE, relatedness, and MAF distributions of the given data. The specific parameters and filtering criteria are detailed below: Filtering missing SNPs: To preserve the integrity and accuracy of the genotype data, the plink --geno parameter was used with a threshold value to remove SNPs with a certain percentage of missing genotypes. A value of 0.02 was determined as the default for PGSXplorer, eliminating variants with more than 2% missing SNP data across the samples (Turner et al., 2011).

Filtering missing individuals: After filtering missing SNPs, individuals with a high rate of missing genotypes are filtered to maintain data quality. In this step of the pipeline, the plink --mind parameter with a value of 0.02 was set as the default value (Turner et al., 2011).

Filtering by MAF: The MAF threshold was determined to filter out rare variants that may introduce noise or bias into the analysis. Generally, values of 0.01 to 0.05 are used for this filtering step (Pavan et al., 2020). In this step, the default value was set to 0.05 using the plink --maf parameter.

Filtering by HWE: Deviations from HWE were evaluated to identify genotyping errors and ensure data quality. In our study, we applied HWE filtering in a two-step process using PLINK. The first filtering step applied a strict HWE threshold of 1e−6 to the control group. This step ensured removal of SNPs that deviated significantly from HWE among controls. We then applied a second HWE threshold of 1e−10 to the case group. This less stringent step only targeted SNPs that showed extreme deviations from HWE in the case data, as the stringent threshold had already been applied to controls (Marees et al., 2018).

Relatedness checking: In population studies, the maximum degree of relatedness between any pair of individuals is typically expected to be less than that of second-degree relatives (Turner et al., 2011). To address this, PGSXplorer identifies and filters possible sample mix-ups or family relationships that may introduce bias downstream analyses. In this step, a default value of 0.1875 was defined for the plink --rel-cutoff parameter.

Heterozygosity assessment: Monitoring heterozygosity levels is essential for identifying potential contamination or issues with genotyping data quality. In the pipeline, heterozygosity filtering is performed based on +/− 3 Standard Deviations (SD) from the mean.

Removal of duplicate SNPs: To identify and remove duplicate SNPs from given genotype dataset, we first listed the duplicate SNPs using the following command: awk ‘{print $2}’ <input_bim_file> | sort | uniq -d > <output_duplicate_snps_list>

This command extracts SNP identifiers from the .bim file, sorts them, identifies duplicates, saves them to a list file. Then PLINK –exclude command was used to remove these duplicate SNPs from dataset. This process ensured that final dataset was free of duplicate SNPs, improving the quality and reliability of our downstream analyses.

Integration of QC, phasing and imputation steps

Following the QC steps of the workflow, the filtered datasets in PLINK format are automatically converted to VCF format for further processing. These VCF files are then phased using Eagle (v2.4.1) (Loh et al., 2016) with reference files tailored to the GRCh38 genome build (details regarding the reference files are available on PGSXplorer’s GitHub page). After phasing, imputation is performed using the same GRCh38-compatible reference datasets with Beagle (v5.4) (Browning, Zhou & Browning, 2018). Subsequent to imputation, additional QC is performed based on imputation scores to ensure data accuracy. The final files are then automatically converted back to PLINK format and made ready for target ancestry inference and PGS calculations.

To include the QC steps, phasing, imputation, and visualizations in the PGSXplorer, the inputs and outputs of each step were defined and assigned to different channels, with separate modules created for each process as shown in Fig. 1. They were carefully designed to filter out low-quality data, thereby enhancing the reliability of the downstream analyses. The initial input, provided in PLINK (bed, bim, fam) or VCF formats, completes the target and GWAS QC steps, phasing, and imputation in an automated and optimized manner. This process prepares the data for PGS calculations. The default parameters used in these steps are determined according to a previous study (Marees et al., 2018), but users can provide the desired values as parameters.

Figure 1 The schema illustrates the integration of QC steps into PGSXplorer.

This workflow applies filters, including checks for missing SNPs, individual genotype quality, MAF, HWE, relatedness, heterozygosity, and duplicate SNPs and illustrations of MAF, HWE, relatedness and heterozygosity distributions. Additionally, it outlines the GWAS QC, phasing, and imputation components of the pipeline.

Target ancestry inference

Detecting ancestry components in genomic data is a critical step in PGS model development. In this study, the Fastmixture (Santander, Martinez & Meisner, 2024) tool was integrated into the pipeline to perform target ancestry inference. Fastmixture utilizes probabilistic modeling approaches to efficiently and accurately determine the proportions of individuals belonging to different ancestral groups by processing genotype data (Santander, Martinez & Meisner, 2024). The resulting outputs of the target ancestry inference step were used in downstream analyses to account for population structure.

PGS modeling of genomic data

After completing the QC steps, PGSXplorer integrates four well-known PGS algorithms —PLINK, PRSice-2, LD-Pred2 (both grid and auto), and Lassosum2 along with MegaPRS and SBayesR-C—to generate robust PGS models from GWAS summary statistics. Each of these tools was chosen for its distinct role and contribution to the development of PGS models. Their selection in this study was based on their widespread recognition and specialized capabilities in the field: (i) PLINK is a well-established tool in genetic association studies, widely recognized for its efficiency in processing large-scale genetic data. Its ability to filter SNPs meeting certain p-value thresholds significantly contributes to PGS calculation (Purcell et al., 2007), (ii) PRSice-2 is highly regarded for its versatility in handling large cohorts, requiring users to have a solid understanding of bioinformatics. It enables the estimation of disease risk based on genetic variants and provides flexibility in PGS model development by allowing adjustment of p-value thresholds to suit specific research goals (Choi & O’Reilly, 2019), and (iii) LD-Pred2 enhances the accuracy of PGS models by incorporating linkage disequilibrium (LD) information, a critical factor in capturing the genetic architecture of complex traits. By leveraging LD, LD-Pred2 improves the predictive power for identifying genetic variants associated with complex traits (Privé, Arbel & Vilhjálmsson, 2020). (iv) Lassosum2 estimates PGS using GWAS summary statistics alone and has demonstrated consistent improvements in prediction accuracy, particularly when modeling multiple PGS derived from various parameters. It is a valuable tool within a reference-standardized framework, especially for its ability to handle high-dimensional genetic data and improve trait prediction (Pain et al., 2020; Privé, Arbel & Vilhjálmsson, 2020). (v) MegaPRS accurately estimates the effect of genetic variants on phenotypic traits by incorporating complex inheritance models. Using the BLD-LDAK model, it considers LD, MAF and functional characteristics of SNPs, which increases accuracy and enables better modeling of genetic variance. Standing out for its computational efficiency, MegaPRS offers an effective and flexible solution for genetic predictions across different populations and traits (Zhang et al., 2021). (vi) SBayesR-C calculates PGS using GWAS summary statistics and LD matrices, modeling genetic effect sparsity through a finite mixture of normal distributions. It handles large-scale genetic data efficiently, shows competitive prediction accuracy compared to methods such as LD-Pred2 and Lassosum, and allows integration with functional annotations to increase its power. This makes it a valuable tool in the field of genetic epidemiology and individualized medicine (Zheng et al., 2024).

Multi-ancestry PGS tools increase the accuracy of PGS models by exploiting genetic variation across different populations, allowing for more precise modeling of allele frequencies and LD patterns (Chen et al., 2014; Ruan et al., 2022). This approach helps overcome the limitations of traditional methods that rely heavily on European-ancestry data and leads to better estimates for non-European populations (Ge et al., 2022; Shim et al., 2023). To improve the accuracy and comprehensiveness of PGS estimates, PGSXplorer includes PRS-CSx and MUSSEL, both of which significantly enhance the pipeline’s capabilities. (vii) PRS-CSx improves PGS predictions by integrating GWAS summary statistics from multiple populations, which is crucial for accurately estimating disease risk across diverse genetic backgrounds. This tool addresses the common problem of reduced predictive power in non-European populations using a cross-population approach (Ruan et al., 2022). (viii) MUSSEL further strengthens the pipeline by applying advanced statistical techniques such as clustering, thresholding, empirical Bayes, which are particularly effective in optimizing PGS across different ancestries (Jin et al., 2023). Together, these tools overcome the limitations of single ancestor models, making the PGSXplorer workflow for diverse global populations, and thus increasing its value in personalized medicine and risk assessment.

Integration of PGS tools

Incorporating PGS models into PGSXplorer required carefully structuring the inputs and outputs for each step, which were subsequently assigned to distinct Nextflow channels to ensure seamless data flow throughout the pipeline. As illustrated in Fig. 2, each modeling process is encapsulated within a separate module, enhancing modularity and flexibility of our pipeline’s design. Tools that model PGS using GWAS data from a single population are categorized as Single PGS, whereas those that incorporate data from at least two distinct populations are defined as Multi PGS.

Figure 2 The schema illustrates the workflow steps integrated into the Nextflow script.

The QC stage is followed by principal component analysis (PCA) to account for population structure and fastmixture. Afterward, optional tools for PGS construction are available. These include multi-ancestry PGS tools such as PRS-CSx and MUSSEL, alongside single-ancestry PGS tools like PLINK, PRSice-2, and LD-Pred2 (available in both auto and grid modes), Lassosum2, MegaPRS and SBayesR-C.

Generation and preparation of synthetic data

To validate the functionality of PGSXplorer, synthetic genotyping data were generated using the HAPNEST (Wharrie et al., 2023). HAPNEST facilitates the creation of synthetic genomic datasets representing various ethnicities, making it ideal for testing and validation purposes. Specifically designed to support genomic research, HAPNEST utilizes containerization through Docker or Singularity, ensuring reproducibility and ease of use. Key features of HAPNEST include the ability to fetch diverse reference datasets customize parameters for specific research needs, and standardize the software environment via containerization.

In our study, the sizes of the synthetic datasets were chosen to present different population sizes to test the performance and efficiency of our pipeline under various conditions. The datasets included 500, 1,000 and 10,000 individuals of European (EUR) origin, and 3,000 and 10,000 individuals of East Asian (EAS) origin. The datasets of 500 EUR, 1,000 EUR, and 3,000 EAS individuals were labeled as T1, T2, and T3, respectively. Using these datasets, both the data processing capacity and computation times of the pipeline were evaluated. Synthetic genomic data were generated for all chromosomes, ensuring comprehensive coverage. Different populations were simulated by modifying polygenicity and genotype proportion values in the HAPNEST configuration file. The commands used to generate these datasets with HAPNEST are as follows:

Genotype data generation: docker run -v /HAPNEST/data:/data -it sophiewharrie/intervene-synthetic-data generate_geno 16 /data/config.yaml

Phenotype data generation: docker run -v /HAPNEST/data:/data -it sophiewharrie/intervene-synthetic-data generate_pheno data/config.yaml

The configuration files used in this process, which detail the parameters and population structures, have been shared on the PGSXplorer GitHub page (https://github.com/tutkuyaras/PGSXplorer), along with the datasets themselves. These synthetic datasets were also utilized to calculate GWAS summary statistics using PLINK2 (Chang et al., 2015). Logistic regression models were employed with the command: plink2 --bfile target --pheno phenotype_file.txt --glm hide-covar --covar target.eigenvec --ci 0.95 --out gwas_sumstat.

Results

All results and figures presented in this study were derived from the analysis performed with PGSXplorer. The figures are intended to provide users with an overview of the tool’s output on synthetic data.

Systematic archiving and visualization of genomic quality control

PGSXplorer is designed to systematically archive the outputs generated at each stage of the QC process in dedicated directories. This approach ensures that the data from all seven QC steps are comprehensively recorded and filtered according to user-defined parameters, enhancing transparency and facilitating downstream analyses. The outputs, ranging from initial data filtration to final QC reports, are clearly organized and readily accessible for further review.

In addition to this structured archiving, the QC module also includes automated graphical representations of key metrics. As shown in Fig. 3, distributions of heterozygosity, HWE, inbreeding coefficients (Pi-hat or IBD), and MAF are visualized and automatically generated for user inspection. QC graph results for T1 and T3 are also given in Figs. S1 and S2, respectively. These visual outputs offer critical insights into the quality of genomic data, enabling rapid identification of potential issues such as population stratification or genotyping errors. By combining progressive data archiving with real-time graphical analysis, the QC process becomes both comprehensive and user-friendly, ensuring high-quality data is prepared for subsequent genomic analyses.

Figure 3 The schema illustrates the graphical analysis of key QC steps in the PGSXplorer pipeline.

The graphs of HWE, pi-hat, heterozygosity rates, MAF distributions, and overall HWE distributions shown in this figure are obtained from the analysis of the PGSXplorer QC module using the target data of the European population of 1,000 individuals (T2) generated with HAPNEST.

The SNP and individual information eliminated according to the parameters used in the QC steps are presented to the user. Table 1 shows the number of SNPs eliminated during QC for the three datasets tested with PGSXplorer. Since a synthetic dataset was used and the GRCh38 rsID (Reference SNP cluster ID) list provided by HAPNEST is common, the initial number of variants is identical. However, the number of eliminated variants and the final number of variants remaining differ across populations. The default parameters of PGSXplorer were applied during QC steps, but users can modify these parameters to suit their specific research objectives and study requirements.

Table 1 Number of remaining SNPs after each filtering steps during the quality control process.

Ancestry	Number of individuals	Initial number of SNPs	Number of remaining SNP after –geno 0.02	Number of remaining SNP after –mind 0.02	Number of remaining SNP after –maf 0.05	Number of remaining SNP after –hwe 10−6	Number of remaining SNP after
– hwe 10−10	Number of remaining SNP after
–pihat 0.185	Number of remaining SNPs after remove duplicates	
EUR (T1)	500	1,329,052	1,329,052	1,329,052	1,041,531	1,041,531	1,041,531	1,041,531	1,023,045	
EUR (T2)	1,000	1,329,052	1,329,052	1,329,052	1,041,708	1,041,708	1,041,708	1,041,708	1,023,224	
EAS (T3)	3,000	1,329,052	1,329,052	1,329,052	949,527	949,527	949,526	949,526	932,520	

Target ancestry inference

Fastmixture, integrated into the pipeline for target ancestry inference, offers an efficient method for analyzing the population structure of genetic data (Santander, Martinez & Meisner, 2024). The Q file generated by this module provides users with ancestry proportions for individuals, indicating the probabilities of each individual belonging to different populations. The .p file, on the other hand, contains allele frequencies specific to each population for genetic variants. This file is particularly useful for examining genetic differences between populations and identifying population-specific genetic patterns.

PGS modeling

The genomic data generated during the QC steps serve as input to the PGS modules, enabling customized analyses. Users can specify which models to execute by using the command nextflow run main.nf–help, ensuring that only the desired modules are and unnecessary computations are avoided. By default, all tools are set to run automatically, except for MUSSEL, which is disabled due to its high computational demands. MUSSEL requires substantial processing power and is designed to be executed only on servers, providing users with the flexibility to opt in for resource-intensive analyses.

As outputs of the PLINK algorithm, the first of the single PGS models, individual-based PGS were generated across seven different p-value thresholds. The resulting scores were stored separately in the “outputs” folder. Table S1 presents the AUC and R² values calculated for the PGS from three different target datasets during the experimentation and optimization phases of PGSXplorer.

The PRSice-2 module generates visual outputs that summarize the performance of PGS models. Figure 4 illustrates the bar plot and high-resolution plot generated based on p-value thresholds and PGS model fit. Along with these visual outputs, the pipeline provides several additional result files, including a list of the best-fitting PGS values determined by the regression model (.best), the number of SNPs used for PGS calculation at each p-value threshold, along with the corresponding R2 and p-values (.prsice), a summary of the model (.summary), and detailed log files. For the LD-Pred2 method, two approaches—auto and grid—have been integrated into the pipeline, offering graphical outputs such as heritability (h2) and p-value plots for the auto model, and GLM-Z score plots for the grid model, as shown in Fig. 5. Additionally, files containing individual principal components and PGS are included in the outputs folder.

Figure 4 Visualizations generated from the PRSice-2 module results.

Bar Plots (A–C) and high-resolution plots (D–F) generated using PRSice-2 for T1 (EUR-500), T2 (EUR-1,000) and T3 (EAS-3,000), respectively. The bar plots (A–C) display the PGS model fit across different p-value thresholds, while the high-resolution plots (D–F) provide a detailed visualization of the model’s performance for the same targets.

Figure 5 Visualizations generated from the LDpred2 auto ve grid models for T1 (EUR-500) and T2 (EUR-1,000).

(A) and (C) display the auto model results for T1 (EUR-500) and T2 (EUR-1,000), respectively, showing h2 and p-value relationships. (B) and (D) present the GLM-Z score plots for the grid model for T1 (EUR-500) and T2 (EUR-1,000), respectively.

Lassosum2, which leverages penalized regression to account for LD, provides an efficient and scalable approach for large datasets. As part of the output, graphical representations, including GLM-Z score graph and post-processed (Postp) plots, are provided to visualize the performance of the Lassosum2 model, as shown in Fig. 6 (the plot for T3 is presented in Fig. S3).

Figure 6 Visualizations generated from the Lassosum2 module.

(A) and (B) display the auto model results for T1 (EUR-500) and T2 (EUR-1,000), respectively, displaying the model fit and performance based on the penalized regression approach for PGS calculation.

MegaPRS and SBayesR-C offer complementary functionalities for PGS modeling, each providing detailed outputs tailored to user needs. MegaPRS supports various statistical models, with BayesR set as the default, allowing users to select the most appropriate model for their data. The outputs include summary files, parameters, correlations, and the model that delivers the best result, all stored in the output folder. Similarly, SBayesR-C generates comprehensive results, including SNP weights and other key metrics for PGS calculation. The output file contains information such as SNP IDs, effective alleles, combined effects at the genotype scale (BETA), probabilities of causation (PIP), and effects at the last iteration (BETAlast).

PRS-CSx, a cross-population PGS modeling approach, calculates PGSs separately for each of the two or more GWAS datasets used. If the --meta parameter is applied, the output includes a meta PGS file that combines SNP effect sizes across populations using inverse-variance weighted meta-analysis of the population-specific posterior effect size estimates. All generated files are saved in the outputs folder. Figure 7 shows the distribution of PRS-CSx module results computed with PGSXplorer, using the T2 (EUR-1,000) and T3 (EAS-3,000) datasets. Figure S4 presents similar results for the T1 (EUR-500) dataset. The integrated MUSSEL tool generates several important outputs. It produces population-specific PGS files with scores calculated for each population based on their genetic datasets and summary statistics, as well as meta-PGS files created using the inverse variance weighted meta-analysis method to combine scores across populations. Additionally, it includes files with SNP-level posterior effect size estimates for each population and meta-analysis results. Configuration files detailing the parameters used in the analysis, such as selected SNPs and population settings, are also provided to the user.

Figure 7 The scatter plots illustrate the distribution of PGS calculated using the PRS-CSx model for chromosome 22.

(A) represents PGS are calculated with T2 (EUR-1,000) and (B) represents PGS are calculated with T3 (EAS-3,000). The red dots represent scores derived from the EAS GWAS data, the blue dots represent scores derived from the EUR GWAS data, and the gray dots show the results of the meta-analysis, which combines both datasets using inverse-variance weighted effect sizes.

Computational performance metrics of PGSXplorer

Computational metrics of the PGSXplorer tool, such as CPU utilization and execution time, are provided to users through Nextflow parameters: -with-report pipeline_report.html and -with-timeline timeline.html. These metrics offer detailed insights into the computational performance of the pipeline. We present these metrics for three synthetic datasets of varying sizes—T1 (EUR-500), T2 (EUR-1,000), and T3 (EAS-3,000)—used to demonstrate the proper functioning of PGSXplorer. Analyses were conducted on chromosomes 1 and 2, and detailed metrics, including runtime and CPU utilization for each dataset, are provided in Table S2. This comprehensive reporting provides users with a clear understanding of the pipeline’s resource requirements across various data sizes and scenarios.

Discussion

Herein, we developed and implemented PGSXplorer, a comprehensive and automated workflow designed to address the challenges associated with calculating PGS from large-scale genomic datasets. The rapid advancement of NGS technologies and the increasing adoption of genomic data-sharing initiatives accelerated individualized medicine efforts (Shi & Wu, 2017). In parallel, GWAS studies, that investigate the relationship between genetic variants and human traits across large populations, have increased in number and size, enabling better identification of genetic factors contributing to complex diseases. Unlike monogenic diseases, where a single gene can be pinpointed as the cause, complex diseases are influenced by multiple genetic and environmental factors, making the concept of PGS pivotal for accurate risk prediction (Choi, Mak & O’Reilly, 2020; Lu et al., 2021).

The accurate calculation of PGS depends heavily on the QC of genomic data, which is one of the most critical steps in any genomic analysis. QC processes are computationally intensive and must handle the large volumes of data generated by different platforms (SNP arrays, NGS, and other high-throughput genotyping technologies) requiring expertise in genomics, statistics, and coding (Zhao et al., 2018). Additionally, combining data from multiple platforms/centers increases the complexity of PGS calculations. Noise in finite GWAS samples and ethical considerations regarding the interpretation of genetic risk further complicate this process. Effective computational methods, such as LD-based clustering and advanced prioritization algorithms, are critical for improving the accuracy of PGS (Pain et al., 2020). Moreover, proper handling of factors such as SNP quality, HWE, relatedness, heterozygosity, and duplicate SNPs are crucial in preventing false-positive results in association studies, underscoring the importance of robust data filtering at every step of genomic analysis.

PGSXplorer integrates several widely used PGS modeling tools, into a streamlined workflow that automates key steps in QC and PGS modeling. This integration not only enhances the accuracy and efficiency of genomic data processing but also allows users to inspect filtered genomic data at each stage, providing flexibility in the analysis. By storing filtered data as separate files at each QC stage, PGSXplorer enables researchers to explore results based on specific filtering criteria, making it easier to refine the data for further downstream analysis. In addition, each PGS tool has been modularly integrated into the pipeline, allowing users to selectively employ only the tools relevant to their specific research objectives. This flexible design optimizes efficiency by avoiding unnecessary computational steps, ensuring that the workflow is tailored to the precise needs of the study, thereby enhancing both time management and resource utilization.

The results generated using synthetic datasets have demonstrated that PGSXplorer is a scalable solution for analyzing genetic risk factors across diverse populations and genomic dataset sizes. One of the most significant strengths of PGSXplorer lies in its support for multi-ancestry genomic analysis through tools such as PRS-CSx and MUSSEL. These tools transcend the limitations of traditional single-population analyses by incorporating genetic data from multiple populations, significantly enhancing the generalizability of PGS models. Furthermore, by facilitating cross-population risk prediction, PGSXplorer offers a more inclusive approach to PGS modeling, which is critical for improving the accuracy of genetic risk prediction across different ethnic groups. This inclusivity ensures that the workflow can be adapted to various genomic studies, providing a valuable tool for researchers aiming to understand the genetic architecture of complex diseases and develop more accurate models. Ultimately, PGSXplorer’s automation and flexibility make it an indispensable tool in advancing PGS research, paving the way for broader applications in individualized medicine across diverse populations.

Currently, PGSXplorer represents a notable advancement over existing pipelines for PGS calculations by providing an open-source, user-friendly solution that is both powerful and accessible. What truly sets PGSXplorer apart is its ability to integrate essential tools such as PLINK, PLINK2, R, Bcftools, Eagle, Beagle, and Python into a single Docker image (tutkuyaras/pgsxplorer_image:v2). This encapsulation ensures that all necessary dependencies are available in the specified versions, eliminating compatibility issues and allowing users to focus solely on their analyses. Users can simply pull this image from Docker Hub using the command “docker pull tutkuyaras/pgsxplorer_image:v2” to have immediate access to the full suite of tools needed for PGS calculations. In addition to simplifying software dependencies, PGSXplorer leverages the power of Nextflow to create a streamlined, reproducible workflow that can be run seamlessly across different computing environments. These flexibilities allow researchers at any skill level to perform complex genomic analyses without needing to invest significant time in setting up software dependencies or environment configurations.

Upon conducting a comprehensive comparison with existing tools, PGSBuilder (Lee et al., 2023) distinguishes itself by supporting six PGS calculation methods (Clumping and Thresholding, Lassosum, LDPred2, GenEpi, PRS-CS, and PRSice-2) and integrating variant annotation functionalities. One of its notable strengths is the web-based interactive interface, which significantly enhances accessibility and ease of use for researchers lacking computational expertise. However, PGSBuilder primarily focuses on single-ancestry data. In contrast, PGSXplorer accommodates both single- and multi-ancestry datasets and offers extensive flexibility by allowing users to customize QC parameters and other workflow steps. This makes PGSXplorer highly adaptable to diverse research needs.

The PGSToolKit (van der Laan, 2018) provides a streamlined workflow through a structured data format and a single configuration file. It supports PGS calculations with PRS-CS, RapidoPGS, and PRSice-2 while including the allelic scoring function of PLINK2. PGSXplorer, however, goes beyond these features by integrating phasing and imputation capabilities, ensuring compatibility with a broader range of datasets. Additionally, PGSXplorer offers a more comprehensive QC process that can be tailored for multi-ancestry data and allows users to modify workflow parameters to suit their specific objectives.

Another recent and significant tool in the field, GenoPred (Pain, Al-Chalabi & Lewis, 2024), is a user-friendly platform designed to facilitate automated and standardized PGS generation. Its ability to handle multiple target file types, support multiple genome assemblies (GRCh36, GRCh37 and GRCh38), perform ancestry inference, calculate scores in the target sample, and generate detailed individual and sample-level reports are key strengths. While our tool currently focuses on binary case-control datasets aligned to GRCh38 in VCF and PLINK formats, it distinguishes itself with automated graphical outputs. These outputs include visualizations for QC metrics such as HWE, MAF, relatedness, and heterozygosity distributions, as well as results from models like PRSice-2, LDpred2, and Lassosum2. Additionally, PGSXplorer integrates multi-ancestry tools, including PRS-CSx and MUSSEL, to enhance accuracy when working with diverse populations.

A noteworthy tool worth discussion is the pgsc_calc (Lambert et al., 2024) pipeline is notable for its ancestry inference module, which matches PGS to relevant populations based on reference datasets. This capability is a strong advantage. However, pgsc_calc relies on preconfigured workflows with limited flexibility for parameter adjustments. PGSXplorer, on the other hand, enables users to produce PGS using nine different tools while offering customizable parameters. This flexibility empowers researchers to tailor analyses to their specific research questions, significantly enhancing the pipeline’s utility and adaptability. A valuable resource, the Michigan Imputation Server (Forer et al., 2024) focuses on improving the interpretability of PGS results through detailed reports and graphical outputs. While this aligns with PGSXplorer’s goal of providing automated visualizations, PGSXplorer further stands out by integrating tools for multi-ancestry analyses. This capability enables PGSXplorer to process datasets from diverse populations, providing broader applicability in genetic research.

In summary, PGSXplorer offers a modular, Nextflow-based architecture that ensures scalability, portability, and reproducibility. By combining robust QC processes, flexible workflows, and automated graphical outputs, it addresses key limitations of existing tools. Its integration of multi-ancestry tools and advanced PGS construction approaches such PRS-CSx and MUSSEL makes PGSXplorer a comprehensive solution for PGS construction and analysis across diverse datasets. Building on this foundation, future updates will further enhance the tool’s capabilities to support a wider range of data formats, study designs, and genomic references. In future updates, our tool will be expanded to include support for additional data formats such as BGEN, in addition to the currently supported VCF and PLINK formats. Currently designed for case-control studies, the tool will also include modules for datasets with continuous traits (such as body mass index (BMI), blood pressure, insulin resistance etc.). Additionally, a validation module will be introduced to enable users to assess the performance of their PGS models using independent datasets where the phenotype of interest has been measured. Our tool, which currently only supports GRCh38, will also be enhanced with an automatic versioning module that ensures compatibility with GRCh37 datasets.

Conclusions

PGSXplorer stands out as a comprehensive and user-friendly tool developed to address the challenges that arise in analyzing large-scale genomic data. By automating complex processes such as the calculation of PGS and QC of genomic data, it increases accuracy while significantly reducing data processing times. By integrating data from different populations, it allows for a more inclusive and generalizable assessment of genetic susceptibility in multi-origin analyses. Additionally, its automation and optimization minimize the complexities associated with QC steps and PGS calculations, particularly for binary traits, making it an indispensable resource for genomic research. By removing many technical barriers inherent in genomic data analysis, PGSXplorer enables researchers to focus on uncovering meaningful insights from their data, thus fostering broader adoption in precision medicine and genetic studies.

By increasing reproducibility and portability through Docker encapsulation, PGSXplorer offers a practical solution for both experienced and new researchers. As detailed in the discussion, future updates are planned to add continuous feature type, different input formats and genomic assemblies, which will further expand the impact and application potential of PGSXplorer in the research field. In conclusion, PGSXplorer is a step forward in genetic research, contributing to more precise and reliable risk assessments in individualized medicine and providing a wide range of applications in genetic research.

Supplemental Information

Supplemental Information 1 Supplemental Figures.

Supplemental Information 2 AUC and R2 values for PGS generated using the Pruning and Thresholding method across seven different p-value thresholds, calculated using synthetic data from chromosomes 1 and 2.

Supplemental Information 3 Analysis completion time and CPU utilization for chromosomes 1 and 2, based on calculations performed using PGSXplorer with T1, T2, and T3 datasets across eight methods (excluding MUSSEL).

The authors would like to thank Prof. Dr. Gül ERGÖR from Dokuz Eylül University for her perspective and support throughout the study and Mr. Hüseyin GÜNER from İzmir Biomedicine and Genome Center for his technical support and Ms. Leman BİNOKAY from İzmir Biomedicine and Genome Center for her support throughout the study. The authors also acknowledge the use of a generative AI tool (ChatGPT by OpenAI) for English grammar checks and minor language editing in the preparation of this manuscript. All scientific content, interpretation, and conclusions were prepared and verified by the authors.

Additional Information and Declarations

Competing Interests

Gökhan Karakülah is an Academic Editor for PeerJ.

Author Contributions

Tutku Yaraş conceived and designed the experiments, performed the experiments, analyzed the data, prepared figures and/or tables, authored or reviewed drafts of the article, and approved the final draft.

Yavuz Oktay conceived and designed the experiments, performed the experiments, analyzed the data, prepared figures and/or tables, authored or reviewed drafts of the article, and approved the final draft.

Gökhan Karakülah conceived and designed the experiments, performed the experiments, analyzed the data, prepared figures and/or tables, authored or reviewed drafts of the article, and approved the final draft.

Data Availability

The following information was supplied regarding data availability:

All data and code generated during the development of the tool are available on the GitHub page of PGSXplorer (https://github.com/tutkuyaras/PGSXplorer) and Zenodo:

tutkuyaras. (2025). tutkuyaras/PGSXplorer: v2.1 (v2.1). Zenodo. https://doi.org/10.5281/zenodo.14637161

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
