# Peer review of "PGSXplorer: an integrated nextflow pipeline for comprehensive quality control and polygenic score model development"

_PeerJ, doi:10.7717/peerj.18973_

## Round 0.1 · original submission · Minor Revisions

The manuscript is well-organized; the background and motivation sections nicely introduce what this study aims to accomplish. As the reviewers mentioned, the Discussion could be strengthened by providing a more detailed comparison with existing alternatives such as PGSBuilder and PGSToolKit, outlining the distinct advantages of PGSXplorer.

Please see the comments of the reviewers and answer the concerns raised by the reviewers.

Reviewer 1 ·

Basic reporting

The manuscript is well-organized and adheres to scientific standards, with a clear flow between sections. The background and motivation for the study are well-articulated, and the rationale behind developing PGSXplorer is presented effectively. However, the Discussion could be strengthened by providing a more detailed comparison with existing alternatives such as PGSBuilder and PGSToolKit, outlining the distinct advantages of PGSXplorer. The manuscript also references key literature in polygenic scoring and quality control, establishing a solid foundation. However, adding recent studies on emerging trends in polygenic risk computation, particularly those focused on diverse population analyses, would provide a more comprehensive literature review.

Experimental design

The research question is well-defined, focusing on addressing the lack of standardized pipelines for polygenic score development, especially those capable of handling multi-ancestry data. The objectives are clear, aiming to streamline the QC and PGS construction processes for large-scale genomic data. The methods are also detailed, with each step of the PGSXplorer pipeline comprehensively described. However, the Materials and Methods section could benefit from additional details on computational requirements at different stages of the workflow, especially for users with limited computational resources. It would also be helpful to clarify any specific parameter adjustments that are recommended for varying population datasets, as this could aid users in achieving optimal results based on ancestry.
The proposed pipeline in the current study is notable for the integration of established tools like PLINK, PRSice-2, LD-Pred2, Lassosum2, PRS-CSx, and MUSSEL within a single Docker-encapsulated Nextflow pipeline. This setup enhances reproducibility, modularity, and ease of use. It may be beneficial to outline each tool’s unique contributions and the reasons behind their selection in more detail, which would further justify their inclusion and value within the pipeline.

Validity of the findings

The results are systematically presented, detailing each QC and PGS step’s impact on data integrity and the effectiveness of the integrated tools for single- and multi-population analyses. The inclusion of synthetic datasets to validate PGSXplorer’s capabilities is a strong point, as it demonstrates the pipeline’s robustness and scalability. However, reporting performance metrics (e.g., processing times or memory requirements across different dataset sizes) would strengthen the case for PGSXplorer’s computational efficiency.
The pipeline is available as an open-source resource on GitHub, which aligns well with FAIR principles. The GitHub repository provides good initial documentation, but additional example workflows, particularly for first-time users, would further enhance accessibility. This could include a walkthrough of commonly used parameters or case studies on specific PGS analyses.
The authors should also mention plans to expand PGSXplorer to support continuous trait analysis and validation modules, which would significantly enhance its utility. Expanding this section with further details on these anticipated updates would provide a clearer picture of the pipeline’s future scope and impact. Additionally, the manuscript would benefit from a more explicit discussion of potential limitations, such as high computational demands for certain tools, and the authors could consider providing guidance on running the pipeline on standard computing environment

Additional comments

Recommendation Decision (Minor Revisions): This manuscript presents a user-friendly pipeline for polygenic score model development, with implications for genomic research and personalized medicine. While it is well-structured and technically sound, a few improvements in the areas outlined above would enhance its clarity, accessibility, and overall impact.

Reviewer 2 ·

Basic reporting

This study introduces PGSXplorer, a Nextflow-based pipeline designed to streamline polygenic score (PGS) development by integrating quality control steps and advanced modeling algorithms. Automated genotype filtering and incorporation of multi-ancestry analyses addresses computational challenges faced during PGS development. Also, integration of data from diverse populations should be expected to yield more inclusive and generalizable assessments of genetic susceptibility. Overall, this tool can be seen as a significant leap forward in facilitating large-scale genomic studies and precision medicine initiatives, offering researchers a scalable solution to derive actionable insights into complex diseases and traits.

Experimental design

Its Docker-based design and adherence to FAIR principles will increase reproducibility and usability, and therefore help make it a valuable tool for large-scale genomic studies and precision medicine research.

Validity of the findings

The methods used are valid and allows other researchers reproduce the results obtained. The manuscript includes appropriated references to the used tools, methods and describes them clearly. Necessary supplementary material is included.

·

Basic reporting

Issues with Table 2: Table 2 contains several errors. For example, commas are used instead of periods to denote decimals, and R² values exceed 1 (e.g., "1,457E+09"). Additionally, the reported AUC values are all around 0.5, which suggests that the PGS from the pipeline may not be achieving meaningful predictive utility. These discrepancies need to be addressed and clarified.

Comparisons and Novelty: The manuscript does not cite existing tools such as GenoPred (https://pubmed.ncbi.nlm.nih.gov/39292536/) or the pgsc_calc pipeline (https://pubmed.ncbi.nlm.nih.gov/38853961/). A comparison with these pipelines is warranted to contextualize the novelty of this work. At present, the pipeline seems to lack unique features and is missing some key components, such as target ancestry inference.

Experimental design

While I commend the authors for their work, I have major concerns that need to be addressed:

Input Formats: The manuscript does not clearly specify the acceptable formats for target sample data and GWAS summary statistics. For target data, is the pipeline restricted to PLINK1 format? If so, this is a limitation since PLINK1 does not handle dosage information and many datasets are stored in other formats such as BGEN, VCF, or PLINK2. For GWAS summary statistics, which are notoriously unstandardized (e.g., variable column names and content like sample size, allele frequency, effect size), does the pipeline accommodate different formats, or does it require users to reformat their data to a specific structure? Additionally, what QC steps are performed on the GWAS summary statistics?

Computing Environment: The pipeline is implemented within a Docker container. Can it leverage high-performance computing (HPC) or cloud computing resources to distribute jobs across multiple nodes or instances? A single-instance Docker implementation may limit scalability for users working with large datasets.

Missing Methods: The pipeline does not appear to include some leading methods, such as MegaPRS and SBayesR-C. These methods are widely used and would provide a more comprehensive set of PGS approaches.

Reference Data and LD Estimation: What reference data is used for PGS methods? Does the pipeline estimate linkage disequilibrium (LD) from the target sample itself? If so, can the pipeline handle data from a single individual? Additionally, how does the pipeline manage GWAS derived from non-European populations? These details are critical for assessing the pipeline's applicability to diverse datasets.

Evaluation of PGS: The manuscript provides no information on how the PGS generated by the pipeline are evaluated. Are performance metrics calculated as part of the pipeline? If so, how are phenotypes specified? If not, what guidance is provided for users to evaluate the PGS?

Handling of Target Sample Ancestry: How does the pipeline address ancestry in the target sample? Ancestry-specific issues are crucial for the accurate calculation and application of PGS.

QC Practices: Many of the target sample QC steps described are typically performed prior to imputation. Is it appropriate to include these steps in the pipeline, given that most datasets are already imputed before PGS calculations?

Validation with Real Data: The pipeline has only been validated using simulated data. To provide a more robust evaluation, the authors should consider testing the pipeline on real data. Publicly available datasets, such as OpenSNP, could be useful for this purpose. Furthermore, the simulated data could be used better to show the PGS from the pipeline perform as expected, but this is not clear currently.

Validity of the findings

Not applicable.

Reviewer 4 ·

Basic reporting

The authors present a novel tool called "PGSXplorer: An integrated next flow pipeline for comprehensive quality control and polygenic score model development." It seems to be very clear and documented. I have some small issues to complement the work:


Figure 3 - MAF distribuion plot and Histogram HWE. This is only one high pick; what should be expected?

Figure resolution must be improved.

- Remove the sentence "Click or tap here to enter text." at line 69 in the introduction.

- Could you include the potential limitations of this work more clearly in the discussion?

- Could authors provide a table with the possible major functionality and analysis for PGSXplorer?

Experimental design

See basic reporting session

Validity of the findings

See basic reporting session

Additional comments

See basic reporting session

---

## Round 0.2 · accepted · Accept

In the previous evaluation, one reviewer already accepted the publication of this manuscript. 1 reviewer has some concerns which is explained in detail. 2 other reviewers requested very few minor revisions.

I would like to thank the authors since they have addressed all of the reviewers' comments during this revision. They provided an extensive response letter including 12 pages. In the revised version they have conducted several new experiments, comparative evaluations with other competitive tools as requested by a reviewer. Based on these new findings, the authors have also edited manuscript extensively. In its current form the manuscript is ready for publication.